# Potential Allergenicity Response to *Moringa oleifera* Leaf Proteins in BALB/c Mice

**DOI:** 10.3390/nu14214700

**Published:** 2022-11-07

**Authors:** Jie Zhang, Xuan Liu, Zhongliang Wang, Hua Zhang, Jinyan Gao, Yong Wu, Xuanyi Meng, Youbao Zhong, Hongbing Chen

**Affiliations:** 1State Key Laboratory of Food Science and Technology, Nanchang University, Nanchang 330047, China; 2School of Food Science and Technology, Nanchang University, Nanchang 330031, China; 3Animal Science and Technology Center, Jiangxi University of Traditional Medicine, Nanchang 330004, China; 4Sino-German Joint Research Institute, Nanchang University, Nanchang 330047, China; 5Jiangxi Province Key Laboratory of Food Allergy, Nanchang University, Nanchang 330047, China

**Keywords:** *Moringa oleifera*, food allergy, potential allergenicity, cytokines

## Abstract

The reported association of *Moringa oleifera* seeds and allergic disease clinically resembling occupational asthma in cosmetic manufacturing workers has resultedin the need to identify such components in the manufacturing process. However, *Moringa oleifera* leaves from the same plant, an important food ingredient, have limited immunotoxicity data. This study aimed to determine if *Moringa oleifera* leafproteins (MLP) can elicit allergic responses in BALB/c mice. The BALB/c mice were sensitized twice and challenged 10 times to evaluate the potential allergenicityof MLP in vivo. The results showed increased levels of mast cells, total and specific IgE and IgG, severe signs of systemic anaphylaxis, and reduced body temperature compared with controls. The sensitized mice serum observed enhanced levels of histamine and Th-related cytokine release. Compared with the control group, increased levels of interleukins IL-4, IL-9, and IL-17A and enhanced expression and secretion of normal T cells were found in the culture supernatant of splenocytes treated with MLP.This study suggeststhat MLPcanelicit allergic responses; this providesmore comprehensive guidance for identifying new allergen candidates and developing hypoallergenic MLP products.

## 1. Introduction

Reports of occupationalasthma in 32-year-old French employeesata manufacturing facility for the cosmetic industry were investigated by scientists from the Department of Pulmonary Function Testing and Exercise Physiology, Centre HospitalierUniversitaire Nancy, Nancy, France [1]. An association was identified between work-related exposure to the powder of *Moringa oleifera*, used as a component of product moisturizers, and occupational asthma(OA), a disease sometimes characterized by cough, chest tightness, dyspnea, and wheezing [2]. IgE responses to *Moringa oleifera* proteins are consistent with a specific IgE immunologic mechanism.High-level exposure to *Moringa oleifera* should be considered a potential cause of IgE-mediated asthma.The *Moringa oleifera* protein is a complex protein that contains many different proteins from 8–70 kD. It has been shown that *Moringa oleifera* seeds have multiple IgE-reactive proteins between 24 and 56 kDa, from immunoblotting with patient serum.

*Moringa oleifera*, better known as the drumstick tree, is native to the foothills of the Himalayan ranges in the Indian subcontinent. It has a variety of medicinal values, such asantioxidant [3], anti-tumor [4], liver protection [5], and other effects. The main use of *Moringa oleifera* is as a food source [6]. The leaves of the *Moringa oleifera* tree are commonly eaten as a vegetable in India, Southeast Asia, and Africa. Mature leaves are not fibrous and are suitable for stir-frying. The leaves can also be dried, ground into a fine powder, and used as a supplement by adding small amounts to soup, bread dough, and stews. It was reported that *Moringaoleifera* leaves contain 25% to 30.3% protein [7]; they havebeen used as an alternative food source to combat malnutrition, especially among children and infants [8]. Sustained consumption of any proteinaceous food, including legumes, may increase the probability of sensitization against the potentially allergenic proteins in susceptible individuals. Food allergy is a risk factor for other allergic diseases. Identifying and treating food allergies can reduce the risk of developing other serious allergic diseases later in life. However, despite the cosmopolitan consumption of *Moringa oleifera*, surprisingly, there are few reports on its allergenicity. Due to its leaf protein content, *Moringa oleifera* is important for tropical malnutrition reduction strategies.

IgE-mediated hypersensitivity reactions are the most common type I hypersensitivity reactions. They refer to hypersensitivity reactions that occur within minutes after the sensitized body is re-exposed to the same antigen. Aside from working asthma, gastrointestinal allergies are the most common protein sensitization. A food allergy is defined as an immune-mediated adverse reaction to food. Up to 6% of young children and 3–4% of adults are now believed to be affected by food allergies [9], and several studies suggest a major increase since the late 1990s. Milk and chicken eggs are the most common food allergens [10]. Diseases caused by food allergies often affect children’s growth and development and may induce other allergic diseases [11]. Because the pathogenesis of food allergies is complex and affected by many factors, such as genetic background, environmental factors, and exposure conditions, establishing animal models to seek new preventive and therapeutic measures is the most commonly used research method.

Previous studies in our laboratory found that *Moringa oleifera* leaf proteins(MLP) is a sensitizer in OVA food allergy reactions. This correlation suggests that, in addition to OA, MLP substances may play a role in allergic disease. There are few studies on the allergenicity of *Moringa oleifera* leaves and their sensitizing allergens. Accordingly, this work evaluates the food-sensitizing potential of MLP and its potential role in the development of allergic disease, as well as itssafety as a food. To establish the ability of the protein to induce an allergic response through a mouse model, we conducted a thorough study of MLP allergy in BALB/c mice and its splenocytes in this paper. The BALB/c mice in this study underwent intraperitoneal sensitization and oral challenge. Clinical signs of the mice were monitored, followed by detection ofserum MLP-specific antibodies, plasma histamine, and splenic cytokine, and morphological changes of the gut were evaluated. This work might provide important information regarding MLP, providing more comprehensive guidance foridentifying new allergencandidates or developing hypoallergenic MLP products.

## 2. Materials and Methods

### 2.1. Materials and Reagents

*Moringa oleifera* leaves were purchased from Zhikang Agricultural Development Co., Ltd. (Yunnan, China). Human salivary a-amylase, DPPH, and DMSO were purchased from Sigma-Aldrich Co. (St. Louis, MO, USA). Methanol, acetonitrile, formic acid, and other solvents and chemical agents were obtained from Merck (Darmstadt, Germany). Water was purified inhouse by a Milli-Q system (Bedford, MA, USA). The superoxide dismutase (SOD), glutathione peroxidase (GSH-Px), total antioxidant capacity (T-AOC), reduced glutathione (GSH), and malondialdehyde (MDA) kits were purchased from Nanjing Jiancheng Bioengineering Company (Nanjing, China).

### 2.2. Extract Preparation

Leaves of *Moringa oleifera* were extracted with 0.15 M NaCl for 6 h at room temperature, resultingin saline extracts from leaves. Proteins of the saline extracts were precipitated using 0–60% ammonium sulfate fractionation for 4 h at room temperature. The 0–60 F was dialyzed with distilled water (two changes) and 0.15 M NaCl overnight [12].

### 2.3. Moringa oleifera Leaf Protein Characterization

#### 2.3.1. SDS-PAGE

MLP was characterized using SDS-PAGE using standard procedures [13]. Briefly, protein samples (5 μg/well) were separated under reducing conditions (2-mercaptoethanol, heated for 5 min at 100 °C) or under non-reducing conditions (without 2-mercaptoethanol and heating) in SDS-PAGE, performed on a Bio-Rad microprotein electrophoresis system (BioRad Laboratories, Inc., Hercules, CA, USA) using 12% separation gel and 5% stacking gel, then stained with colloidal blue G250 (Sigma, Saint Louis, MO, USA) [14]. The gel was then scanned with the Kodak Gel Logic 440 imaging system, and the density of pixels of the primary band at the appropriate molecular weight in reducing conditions was compared to the total density of all protein bands in the sample lane.

#### 2.3.2. Fourier Transform Infrared (FT-IR) Spectroscopy

The infrared spectra of the MLP study used an FTIR spectrometer (Nicolet 5700, Thermo Nicolet, Waltham, MA, USA) with KBr pellets [15].

#### 2.3.3. Amino Acid Analyze

The protein was prepared into an aqueous solution of 0.1 mg/mL and then entered into an amino acid analyzer (L-8900, Hitachi, Tokyo, Japan) for analysis, and each amino acid peak of the sample was qualitatively and quantitatively analyzedwith an external standard [16,17].

#### 2.3.4. MALDI−TOF MS Sample Preparation and Data Acquisition

MS samples were prepared according to the Shimadzu manual (dried-droplet crystallization method; Axima^®^ Performance Start Guide). Briefly, the matrix solution was prepared by dissolving 5 mg of MLP into 0.5 mL of acetonitrile/0.1% trifluoroacetate (Sigma-Aldrich, St. Louis, MO, USA) 1:1 solution in a clean centrifuge tube, followed by centrifugation at 20,000× *g* for 5 min. The supernatant was carefully mixed by pipetting with an equi-volume of the purified protein (analyte).

For MS detection, 1 μL of the mixed sample was applied onto a sample plate (Shimadzu, Kyoto, Japan) and allowed to dry completely under ambient conditions. An AXIMA Performance mass spectrometer (Shimadzu, Kyoto, Japan) was used in this study. Respective MS data acquisition was performed under the following conditions: power, 90; profiles, 50; shots, 1000; linear mode; with pulsed extraction optimized at 66,430.09 Da, equivalent to the molecular mass of BSA (Sigma, St. Louis, MO, USA) used as the calibration standard [18].

### 2.4. Sensitization Protocol of BALB/c Mice

#### 2.4.1. Animals

In this study, eight-week-old female BALB/c mice, with abody weight of 20–25 g, were purchased from Hunan STA laboratory animal CO., Ltd. (Changsha, China; permission number “SCXK(XIANG)2019-0004”). The animals were housed at a maximum of 5 per cage in ventilated plastic shoebox cages with aspen shaving bedding, 6% irradiated rodent diet (not *Moringa oleifera* leaves), and pure water provided from water bottles adlibitum. The animals were kept at atemperaturebetween 20 °C and 26 °C, 12-h light/dark cycle, and relative humidity between 40% and 70% (Jiangxi University of Chinese medicine, laboratory animal research center for science and technology, Animal House; permission number “SYXK(GAN)2017-0004”). All the experimental manipulations followed the guidelines of the care and use of experimental animals in the Faculty and were approved bythe Animal Experiment Committee (Approval code JZLLSC20210087).

#### 2.4.2. Experimental Design

The animal experiment is depicted in Figure 1 [19]. However, the dose was adjusted because the sensitizer differed from the reference. For example, because MLP is a mixed protein, part of which may be the allergen, the sensitization amount was 6–12 times higher than the 50 μg of OVA, and the excitation amount was set at 5 mg due to the solubility. Briefly, mice were randomly divided into three groups: Group I, a control (CT) group; Group II, MLPlow-dose sensitization and challenge group (MLP Low-dose); Group III, MLP high-dose sensitization and challenge group (MLP High-dose).

Mice in the MLP low-dose and MLP high-dose groups were sensitized, i.p., with 300 μg/600 μg of MLP in 0.2 mL of phosphate-buffered saline (PBS) and 1 mg of aluminum hydroxide adjuvant on day 0 and day 14; two weeks later, they were orally challenged with 5 mg MLP in 0.2 mL saline at 2–3 day intervals to establish a mouse model of food allergy. Mice in the control groups were sensitized with 1 mg of aluminum hydroxide adjuvant and challenged with saline. On the challenge day, the treatment was performed 1 h before the challenge. Diarrhea was assessed visually by monitoring mice for 30 min after each i.g. exposure. Body weight was recorded before gavage and rectal temperature before and 30 min after each i.g. exposure.

#### 2.4.3. Assessment of the Sensitizing Capacity of MLP

We studied the clinical symptoms to assess the sensitizing capacityof allergic reactions induced after the challenge with a heavy amount of proteins in sensitized mice. Thepotency of the food allergy was determined by assessing the rectal temperature change, the systemic anaphylaxis symptoms score, and the diarrhea score. The rectal temperatures were measured just before and 1 hafter the last challenge using adigital thermometer (Beurer GmbH, Uttenweiler, Germany). However, the systemic anaphylaxis symptoms score was evaluated immediately after the last challenge, where each mouse was placed in an individual cage and monitored for 1 h in a double-blind method. Then, the systemic score (0–6) was defined as the sum of scores obtained from three criteria: scratching behavior, loss of mobility, and swelling. Scratching was graded based on the average number of scratching events during each 15-min interval (three intervals during a 45-min observation) as follows: 0, 1–3 events; 1, 4–5 events; 2, >6 events. Loss of mobility was graded according to immobility duration as follows: 0, <10 min; 1, >10 min; 2, during the entire trial. Swelling (bristled fur, edema around the nose and eyes, and laborious breathing) was graded as follows: 0, none; 2 swelling [20]. Diarrhea was assessed by visually monitoring mice for up to 1 h following the intragastricchallenge. Diarrhea was arbitrarily scored as follows: 0, no fecal changes; 1, soft but well-formed feces; 2, soft and non-formed feces; 3, oneepisode of liquid diarrhea; 4, at least two episodes of liquiddiarrhea; 5, score 4 plus only clear liquid in the colon at thesacrifice [21].

#### 2.4.4. Estimation of Specific IgE and IgG1 Levels

Because of the pivotal role of IgE in allergic reactions, we sought to estimate the MLP-specific IgE level in the sera of MLP-treated mice. Specific IgE and IgG1 levels were estimated according to previously described methods with slight modifications [20]. Briefly, MLP wascoated on flat-bottomed 96-well plates using 2 μg protein per ml in sodium bicarbonate buffer, applied at 100 μL/well. Plates were sealed and incubated overnight at 4 °C. After cleaning, plates were blocked for 1 h at room temperature using a 200 μL/well blocking agent. Sera were diluted 1:10 using assay diluent and incubated for 1 h at room temperature before being pipetted into predetermined wells in the washed ELISA plate. Mouse serum was tested in duplicate wells. Plates were then sealed and incubated overnight at 4 °C. Plates were washed and soaked five times with 60 s soak-times with 250 μL/wellwashing buffer, and bound IgE was detected using HRP-conjugated anti-mouse IgE (1:200, *v*:*v*) (BD Biosciences-PharMingen, San Jose, USA) and HRP-conjugated anti-mouse IgG1 (1:1000, *v*:*v*) (BD Biosciences-PharMingen San Jose, CA, USA), added at 100 μL/well. Plates were sealed and incubated for 1 h at room temperature. BD OptEIA TMB substrate reagent set (BD eBioscience, San Jose, CA, USA) was added at 100 μL/well and wasdeveloped for 30 min at room temperature. Finally, ODs were read at 405 nm (ThermoVarioskan Flash, XBFZSW-0008, Waltham, MA, USA).

#### 2.4.5. Assessment of Serum Histamine and Cytokine Levels

The serum histamine, IL-4,IL-17A, IL-13, IL-23, and IFN-γ levels were measured by double-antibody sandwich enzyme-linked immunosorbent assay kits manufactured by SunRed Bio Tech., Shanghai, China, according to the manufacturer’s instructions, with minimum detection limits of 6.5 pg/mL for histamine, 0.1 pg/mL for IL-17A, and 1 pg/mL for IL-4, IL-13, IL-23, and IFN-γ. All optical densities were analyzed at 450 nm using an automated ELISA plate reader, Stat Fax-2100, Fisher Bio-block Scientific, France, and expressed as pg/mL [22].

#### 2.4.6. Cell Culture and Cytokine Evaluation

Mesenteric lymph nodes (MLN) and spleens were removed at sacrifice. Single-cell suspensions were prepared in RPMI-1640 containing 10% fetal bovine serum (BioClot GmbH, Aidenbach, Germany) and 1% Antibiotic-Antimycotic solution (Sigma-Aldrich). Cells (6 × 10^5^/well) were cultured in a flat-bottom 96-well plate without any stimuli or in the presence of MLP (100 mg/well) for 72 h (37 °C, 5% CO_2_) [23]. Supernatants were collected and stored at −80 °C until analyses. IL-4, IL-17A, and IL-9 were determined [24] as 2.4.5. Values are reported in pg/mL after subtracting baseline levels of non-stimulated cultures. Values below assay sensitivity were considered non-detectable (n.d.).

#### 2.4.7. Histology and Morphometry

After the mice were sacrificed and after immersion in disinfectant alcohol for 5 min, the jejunum, ileum, and colon tissues of the mice were separated in a sterile operating environment and placed in a fixative solution (for maldehyde solution containing 4% phosphate) for preservation. Intestinal tissue sections were fixed immediately in 4% formalin. The fixed tissues were cut and processed using routine methods. Paraffin sections (5 mm) were deparaffinized in xylene, rehydrated through an ethanol gradient to water, and stained by hematoxylin and eosin. Paraffin sections (5 mm) were stained by Toluidine Blue Staintoobserve the mast cells. The mast cells were examined and counted under a light microscope (Olympus, Japan) [21].

### 2.5. Statistical Analysis

Results were expressed as means ± SD. All data were analyzed by the SPSS statistical software, version 19.0 (SPSS Inc, Chicago, IL, USA), using one-way analysis of variance (ANOVA), followed by Duncan’s multiple range tests to estimate statistical significance at the level of *p* < 0.05. GraphPad 7.0 software (San Diego, CA, USA) was used for graph drawing and statistical analysis.

## 3. Results

### 3.1. MLP Conforms to the Characteristics of an Allergen

Aspects of the protein structure likely to be relevant for allergenicity are solubility, stability, size, and the compactness of the overall fold [25]. An allergen must therefore contain at least two IgE binding sites (epitopes), each of which will be a minimum of approximately 15 amino acid residues longsoantibody binding can occur.

To preliminarily evaluate the molecular weight of MLP, tricine SDS−PAGE was performed. As shown in Figure 2A, MLP showed six obvious bands. One type of protein was mainly distributed between 35 kDa molecular weights, and other types of protein were 15, 40–50, 50–70,70–100,and 100–120 kDa molecular weights, consistent with the molecular weight of an allergen protein [26]. To identify and obtain the preliminary properties of MLP, matrix-assisted laser desorption/ionization time-of-flight mass spectrometry (MALDI-TOF-MS) was used [27]. As shown in Figure 2B, the molecular weight of the MLP peptide was mainly in the range of 400–1200 Da and 12–19, 24–27, and 48–55 kDa.

Infrared spectroscopy FT-IR is a common method used to identify the structure of biological macromolecules. Through the penetration of infrared light, each chemical bond or functional group of the molecule in the sample to be tested will correspondingly form different absorption and vibration frequencies. Therefore, the molecule’s chemical bond or functional group can be judged according to the formed absorption peak [28]. It was found that MLP exhibited characteristic absorption of protein compounds at 4000–400 cm^−1^, and strong absorption peaks appeared around 3420–3440 cm^−1^ (Figure 2C), which are typically due to N-H stretching vibration. The characteristic peak of the amide A band formed; the absorption peak at 2929 cm^−1^ was the characteristic peak of the amide B band generated by C-N stretching vibration. There wasa large number of peptide bonds between amino acids. The stretching vibration of C=O led to the characteristic absorption peak of amide I at 1700~1600 cm^−1^, and there wereabsorption peaks at 1660~1640 cm^−1^, of which the absorption peak at 1636 cm^−1^ belonged to β-sheet, while at 1654 and 1664 cm^−1^, both had α-helix structures. The absorption peak at 1450 cm^−1^ was a characteristic of the amide II band caused by the C-N stretching vibration. Like the amide I band, the amide II band was composed of α-helix, β-helix, β-turn, and random coil. The combined action of superposition produces the absorption bands. The carbonyl group’s stretching vibration caused the existence of amide I and amide II in the polypeptide, which was related to the degree of cross-linking between the peptide chains. The higher the vibration frequency, the tighter the binding of the peptide chains. The absorption peaks at 1384 and 1383 cm^−1^ were amide III bands caused by C-N stretching vibration. The existence of amide III confirmed the integrity of the protein triple helix structure.

Amino acid composition analysis is the gold standard for accurately quantifying peptides, proteins, antibodies, and other samples. It was found that MLP was composed of twenty-two amino acids, includingfour essential amino acids and eighteen nonessential amino acids. The main amino acids were glutamic acid (acidic amino acids), glycine (fatty acids), and arginine (basic amino acid) (Table 1).

### 3.2. The Ability of MLP to Induce an Allergic Response in Mice

Allergenic evaluation of allergens typically uses in vitro serological or cytological and in vivo validation. Serum is provided by a patient allergic to a given food, and protein is extracted from the food. Food-extracted proteins are separated into protein bands by SDS polyacrylamide gel, and the bands are identified and recognized by IgE antibodies in serum. After separating the bands, they are sequenced to identify the sensitizing protein. Since there are few studies on MLP allergy, obtaining serum in clinical practice is difficult. So, our studyis the first to establish an MLP sensitization mouse model to conduct a preliminary evaluation of the sensitization of MLP. Animal models are an important biological tool to assess the allergenicity of foods or proteins. Some studies have shown that BALB/c mice orally challenged with food allergens exhibit clinical allergic symptoms, such as nose and mouth scratching, reduced activity, shivering, diarrhea, and decreased body weight [29]. In this work, the clinical signs and changes in rectal temperature were observed. The findings indicated that the mice in the MLP group showed anaphylactic reactions from mild to lethal. In contrast, the control groups exhibited minor anaphylactic symptoms (Figure 3A). Compared to the control group, the MLPgroup exhibited a significant elevation of the diarrheal score (Figure 3B). Hypothermic response and allergic symptoms in mice revealed that MLP mightbe potentially sensitizing, like other major allergens (OVA [30], milk protein [31], and peanut protein [32]).

In addition, allergic sensitization to proteins involves the induction of an IgE response of sufficient magnitude to elicit an inflammatory reaction following subsequent exposure to the same (or a cross-reactive) allergen. Sensitization occurs when these IgE antibodies are distributed systemically and bind to FcεR on mast cells and basophils [33]. In mice, the classic pathways that lead to anaphylaxis aremediated by IgE, Fcε receptor I (FcεRI), mast cells, histamine, and platelet-activating factor (PAF) [34]. Food allergy can lead to immunization and the production of specific defensive IgG antibodies [35]. The MLP-sensitized and orally challenged mice significantly increased the MLP-specific IgE and IgG (Figure 3D,E). It is worth noting that the content of IgE is not proportional to the dose of MLP, and clinically, the level of IgE is not consistent with the severity of the food allergy, which may be related to the allergen response threshold of individual mice. The results were generally consistent with the allergic symptom score and rectal temperature results, indicating that MLP mainly caused an allergic reaction mediated by IgE. Any histamine released throughout the body may cause a ‘systemic’ reaction (such as anaphylaxis), so the enhanced release of different allergic mediators like histamine plays an important role in food allergy reactions [36]. Moreover, histamine in MLP low-dose and high-dose groupswassignificantly higher thanin the control group (Figure 3F). The released histamine affected H1 and H2 receptors, resulting in the contraction of smooth muscles of the gastrointestinal and airway (GI) tract, enhancing vaso-permeability and vasodilation and elevating mucus production, cutaneous vasodilation, pruritus, and gastric acid secretion. Therefore, histamine in the gut causes abdominal pain, vomiting, and diarrhea, which are symptoms of food allergy [33,37].

The mast cell plays a very important role in allergic onset via the release of several allergic mediators. By detecting the number of mast cells in jejunum tissue, the characteristics of intestinal pathological changes can be intuitively revealed [38]. Mast cells in the MLP group were significantly increased compared with the control group (Figure 3G,H), which indicated the ability of MLP to induce intestinal allergic responses.

### 3.3. Activated T Cells Strongly In Vivo

Type 2 immune responses elicited by food allergens, including TH2 cytokine responses and IgE antibody production, are key immunological features of allergic disease. TH2-type CD4^+^ T cells and the cytokines that produce IL-4, IL-5, and IL-13 are considered major players in the pathophysiology of allergic diseases. MLP induced a large differentiation of CD4^+^ T cells, affecting the changes of cytokines released by different T cell subtypes. Therefore, we used the cytokine detection kit to detect the Th1 cytokine (IFN-γ), Th2 cytokine (IL-4 and IL-13), and Th17 cytokine (IL-17A and IL-23) change levels.

#### 3.3.1. Effects of MLP on Th1 and Th2 Cytokine Secretion

It is generally believed that food allergy is related to the imbalance between Th1 and Th2 cells, and recent evidence was accumulated to suggest that allergen-reactive Th2 cells triggerallergic inflammation [39,40]. Most food allergen–specific CD4^+^ T lymphocytes were found to synthesize high levels of IL-4 and IL-13, two cytokines required for initiating IgE synthesis [41]. As shown in Figure 4A,B, IL-4, and IL-13 were significantly increased in the MLP group, consistent with the literature, which reportsthat immune cells from food-allergic children up-regulated Th1 and Th2 cytokine secretion induced by food allergens [42].

#### 3.3.2. Effects of MLP on Th17 Cytokine Secretion

In addition to Th1- and Th2-related cytokines, we also explored changes in Th17-related cytokines, a novel subset of cells. The changes in the content of Th17-related cytokines and the changes in cytokines secreted by Th2 cells were similar (Figure 4B); the IL-17A and IL-23 content of MLP-sensitized and stimulated mice was increased significantly when compared with control mice. The specific role of Th17 cell response in food allergy is still unclear. However, in the pathogenesis of asthma, Th17 can aggravate asthma to a certain extent [43]. Since the first case of *Moringa oleifera* allergy was also occupational asthma, we speculated that Th17 cells mightbe related to the dyspnea score exhibited by *Moringa oleifera*–challenged mice.

### 3.4. Activated T Cells Strongly in Sensitized Mouse Spleen Primary Cells

To further explain the enhanced IgE and Th2-type responses, we examined MLP-specific cellular responses. We isolated splenocytes from MLP-sensitized mice and measured the changes of Th2-, Th9-, and Th17-related cytokines in the cell supernatant after MLP stimulation. IL-4, IL-17A, and IL-9 were increased significantly in the MLP group (Figure 5), which wassimilar tothe changes in vivo (Figure 4). The inflammatory response factors (IL-4 and IL-17A) were significantly increased; IL-4 is critical for the polarization of Th2 cells and IgEclass-switching in B cells, suggesting that stimulated MLP primes T cells for a typical TH2 cell type response. This result again verified the allergenicity of MLP. In addition to IL4 and IL17, IgE-mediated intestinal pathological changes in food allergy are driven by IL-9. Previous studies have revealed that IL-9+ MMC9s (IL-9-producing mucosal mast cells) were a key step in IgE-mediated food allergy susceptibility [44]. Nevertheless, the role of Th9 cells, which secrete IL-9, in this process is unclear. In this experiment, MLP re-stimulation in vitro significantly up-regulated the level of IL-9 cytokine, which revealed that IL-9–producing cells in splenocytes mightbe the focus of future attention.

### 3.5. Histopathological Changes of Mouse Gut Tissue

The body produces a series of allergic symptoms when an allergic reaction occurs, such as allergic dermatitis, diarrhea, asthma, and even death [45]. Maintaining the integrity and function of the mucosal epithelial barrier is important for maintaining intestinal homeostasis [46,47]. Research has shown a link between the development of a food allergy and changes in gut barrier function [48]. Histomorphology remains a powerful routine method for assessing intestinal inflammation in animal models [49]. To study the tissue changes in mice with allergic reactions after being challenged, we selected the most possibly affected tissue (intestine) in the MLP allergic reaction for tissue sectioning and observed its pathological changes under a microscope. Figure 6A–C reflect the changes inmouse jejunum, ileum, and colon tissues, respectively. The intestinal villi of the control mice were regular, and the intestinal tissues of the mice sensitized and stimulated by MLP began to develop lesions. In the low-dose MLP group, some glands of the jejunum were arranged disorderly, the villi were flat, and some villi were broken, accompanied by inflammatory cell infiltration (mainly lymphocytes). Most villi were destroyed and disappeared, and the glands were irregularly arranged. In the ileum, part of the glands in the low-dose MLP group were in disorder and the villi were broken and disappeared, and in the MLP high-dose group, a large number of inflammatory cells infiltrated (mainly lymphocytes). A small part of the villi disappeared, and the irregular arrangement of glands was disordered. The colon tissue remainedunchanged. In short, from the above intestinal tissue slices, MLP sensitized and stimulated severe jejunum and ileum lesions in mice, which is related to the above allergic symptom scores, changes in body fluid–specific antibodies, and changes in cytokines released by cellular immunity; the results obtained from the expression of effector cells are consistent. Moreover, it was shown that the allergic reaction mainly acted on the jejunum and ileum and had no effect on the colon.

## 4. Discussion

As an important source of protein supplement, *Moringa oleifera* leaves have been introducedor importedto many countries that did not previously grow and eat *Moringa oleifera* leaves because of their extremely high protein content and nutrients [6]. Almost all edible plants have been reported to cause allergies [50]. As a food rich in plant protein, MLP may contain allergen protein. In general, many allergens in plants are mainly composed of proteins with low molecular weight and disulfide bonds. According to these properties, we determined the molecular weight, three-dimensional structure, and composition of the extracted MLP.SDS-PAGE gel electrophoresis is mainly used to determine the molecular weight of allergens [51]. However, the relative mobility of some proteins in the SDS system does not show a linear relationship with their molecular weights, so SDS-PAGE shows errors in determining molecular protein weights. To more accurately determine the molecular weight of MLP, we also used the MALDI-TOF-MS method to accurately determine the molecular weight of protein subunits from *Moringa oleifera* leaves, as shown in Figure 2. To better understand the properties of MLP used in this study, the protein structure and amino acid composition were identified by infrared spectroscopy and amino acid analysis. The structural information of these proteins’ molecular weight can also help to further the functional identification of leaf proteins. This result will provide some basic information for a better understanding of the potential allergens of MLP.

Although there have been clinical literature reports of allergic reactions to *Moringa oleifera* seeds, there is no report of allergic reactions to *Moringa oleifera* leavesnor experimental studies on this allergy. In vivo evaluation of laboratory animals is a very reliable method for sensitization assessment [52,53]. We hypothesized that MLPs are potential food allergens. We tested this hypothesis by intestinal sensitization of BALB/c mice with MLP followed by gavage challenge. Allergic reactions were detected by noninvasively measuring body temperature and observing symptoms (Figure 3A,B). Our results showed that BALB/c mice were highly sensitive to MLP, developed specific IgE antibodies and Th2 responses, and dramatically decreased core body temperature. Wealso demonstrated significant intestinal mast cell expansion, enhanced intestinal permeability, and the ability to generate a strong allergic response when challenged in the gut. Mast cells were significantly increased in the jejunum of MLP-challenged BALB/c mice, suggesting that mast cell activation drives the physiological response. The intestinal challenge of wild-type BALB/c mice sensitized by this protocol did not elicit significant responses.As PingTong and colleagues recently described [54], similar responses were elicited in BALB/c mice sensitized with known allergens OVA or α-lactalbumin [55]. Asarea and others used ultra-supplementation of *Moringa oleifera* leaf extract and found that high-dose (3000 mg/kg body weight) ultra-supplemented animals did not die and did not have liver and kidney toxicity, and the hematological results were within the normal range [56]. This result also suggested that the disease symptoms caused by MLP in mice were due to allergic reactions rather than toxicity.

Serum total IgE and IgG were significantly increased, especially IgG. IgG antibodies can efficiently capture allergens before reaching cell-bound IgE [35,57], downregulate IgE receptor FcεRI signaling, and promote IgE internalization in mast cells [33]. It is suggested that the allergic reaction caused by MLP may be a food allergy mediated by IgE or IgG. IgE-mediated hypersensitivity reactions are usually I-type, involving an imbalance between Th1 and Th2 responses [58]. Th2-related cytokines were significantly increased in this experiment, but the level of Th1 cytokines was also significantly increased (Figure 4A,B), which could antagonize the differentiation and activation of Th2 cells [59]. In this experiment, obvious intestinal mastocytosis and the disappearance of intestinal villi destruction, irregular arrangement of glands, and increased permeability may be related to IL-4 promoting allergic sensitization to ingested antigens [60], impairing tolerance, supporting intestinal mast cell expansion and drivingIgE-dependent allergic responses. Mast cells produce type 2 cytokines and TH2 chemokines and may recruit TH2 cells to the gut [60].

Previous studies in humans and mice had identified central roles for IL-4 and IL-13 in the pathogenesis of gastrointestinal allergic reactions [61]. According to reports, genetic polymorphisms affecting the IL-4/IL-13 axis were associated with asthma susceptibility [62] and risk association between IL-4 and IL-13 genes polymorphisms and food allergy [63]. The findings suggest that the MLP food allergic response may be driven by enhanced IL-4 signaling, leading to increased Th2 induction and IgE production.

It has recently been shown that IL-9, produced by epithelial cells and other cells, is a significant mast cell growth and differentiation factor and is a pleiotropic cytokine involved in Th2 inflammation in intestinal allergic responses [64,65]. Forbes and colleagues demonstrated that IL-9–driven expansion of intestinal mast cells results in enhanced intestinal permeability [44]. Elizabeth E. Forbes et al. found that an IL-9-stimulated mast cell-mediated increase in intestinal permeability is central to the induction of oral antigen sensitization [66]. Recruitment of mast cell progenitors to allergen-challenged lungs in sensitized mice is IL-9–dependent [67]. In our study, a significant increase in jejunal IL-9 levels was observed in the supernatant of spleen cell cultures from mice (Figure 5), suggesting a promoting role of the spleen epithelium on mast cell accumulation. This finding further confirmed that non-hematopoietic cells also contribute tomast cell accumulation, although susceptibility to allergic reactions is primarily mediated by hematopoietic cells such as mast cells.

## 5. Conclusions

In this study, we aimed to evaluate the sensitization potential of the MLP. To that end, we first developed a clinically relevant mouse model of allergic diarrhea, for which IgEwasthe main mediator and in which challenge wasperformed by oral administration of allergens, similar to what occurs in humans.The increased levels of specific IgE/IgG, histamine, mast cells, and Th1-, Th2-, and Th17-related cytokine cells, symptoms of systemic anaphylaxis, reduced body temperature, and histopathological changes validated the sensitivity to MLP in BALB/c mice. Moreover, the release of IL-4, IL-17A, and IL-9 in the primed sensitized mouse spleen cells after MLP exposure confirmed the *Moringa oleifera* leafallergy in vitro. Our study showed that MLPmighthave the ability to induce allergic responses in vivo and in vitro conditions.

One of the limitations of this experiment is that interlaboratory reproducibility may also be required to use predictive assays for protein risk assessment more reliably. On the other hand, future work will involve clinicians to determine the allergenic proteins by a skin prick test (SPT) or isolating pure proteins and performing immunoblotting with the serum of allergic patients. Changes in gut microbial composition and metabolic activity can affect all aspects of the mucosa’s innate and adaptive immune processes [68]. Therefore, studying gut microbes is also an important direction for further research.

## Figures and Tables

**Figure 1 nutrients-14-04700-f001:**
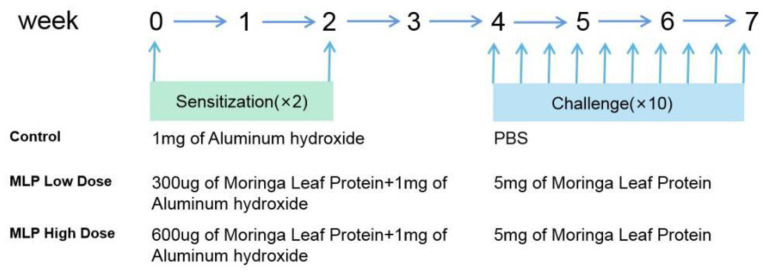
Mouse immunization protocol. Mice were sensitized with two intragastric administrations of MLP (MLP, 30 μg/60 μg per mouse) at a 14-day interval (days 0, 14) and challenged with MLP at a 2-day intervals (days 28, 30, 32, 34, 36, 38, 40, 42, 44, and 48) to establish a mouse model of food allergy in the MLP group or phosphate-buffered saline (PBS) in the controlgroup.

**Figure 2 nutrients-14-04700-f002:**
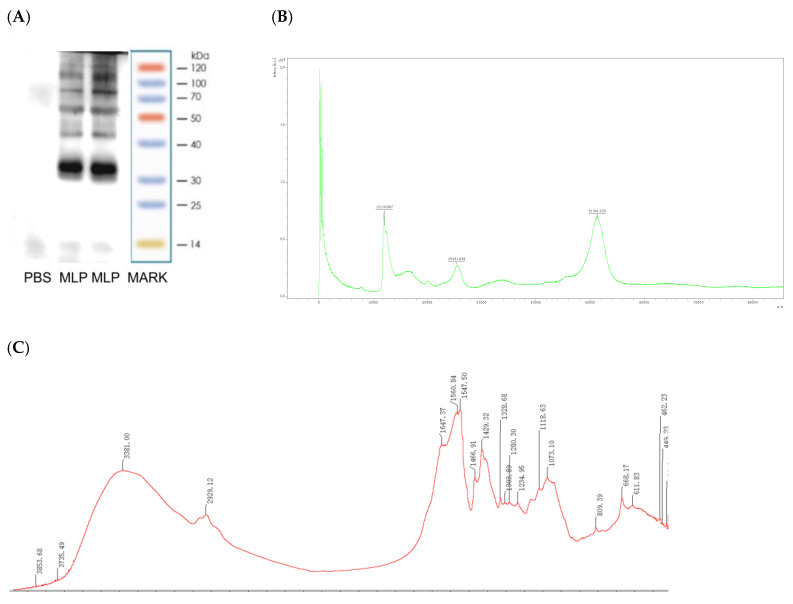
SDS-PAGE (**A**), MALDI-TOF MS (**B**), and Fourier transform infrared spectroscopy spectra (**C**) for MLP.

**Figure 3 nutrients-14-04700-f003:**
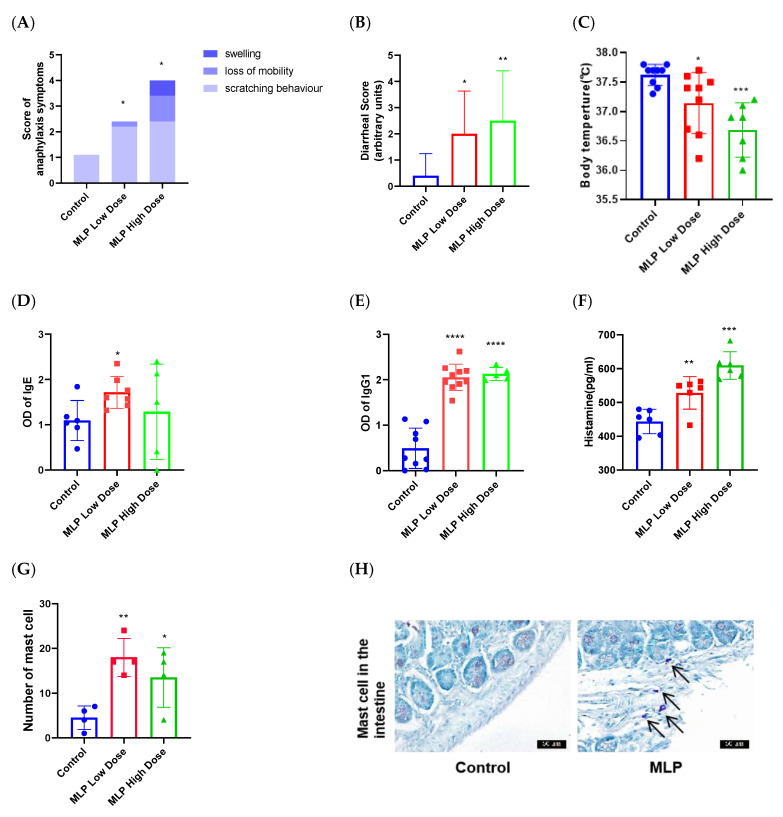
The score of hypersensitivity symptoms (**A**), diarrhea (**B**), and rectal temperature (**C**) in the MLP low-dose group and MLP high-dose group. Levels of specific antibodies (IgG or IgE) ((**D**,**E**), fix the background) and histamine (**F**) in sera from mice exposed to low-dose MLP, high-dose MLP, or PBS (control group) were detected by ELISA. The number of mast cells (**G**) in jejunum following MLP and saline challenges were assessed by morphometric analysis of toluidine blue-stained cells (*n* = 4 mice per group). Histopathological analysis of jejunum ((**H**), arrows point at mast cells). Each point representsdata from an individual mouse, and values are expressed as means ± SD (*n* ≥ 6). * *p* < 0.05, ** *p* < 0.01, *** *p* < 0.001,**** *p* < 0.0001 by two-tailed unpaired Student’s *t*-test where indicated or Mann–Whitney U test.

**Figure 4 nutrients-14-04700-f004:**
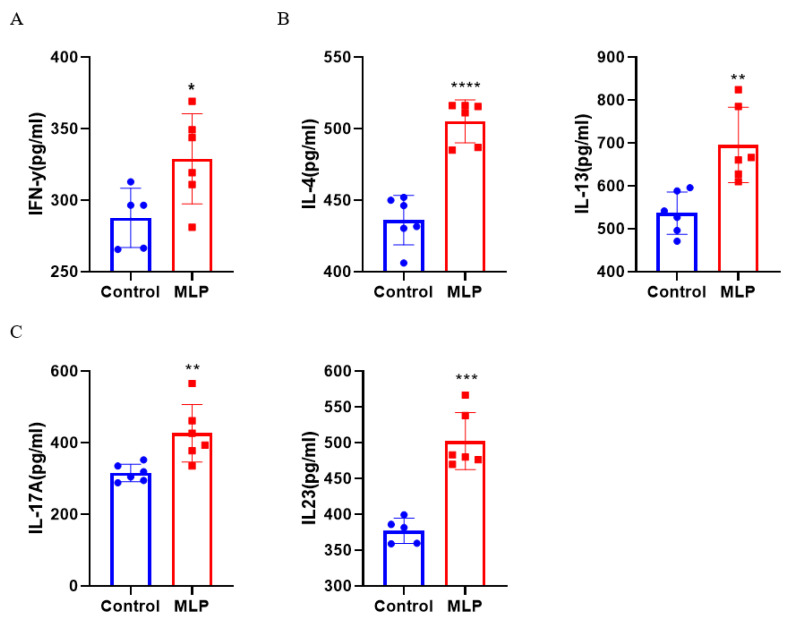
(**A**) Level of Th1 (IFN-γ), (**B**) Th2 (IL-4 and IL-13), and (**C**) Th17 (IL-17A and IL-23) in MLP-sensitized mice sera by ELISA. Significant difference (* *p* < 0.05, ** *p* < 0.01, *** *p* < 0.001, **** *p* < 0.0001; *n* ≥ 5) compared with the control group.

**Figure 5 nutrients-14-04700-f005:**
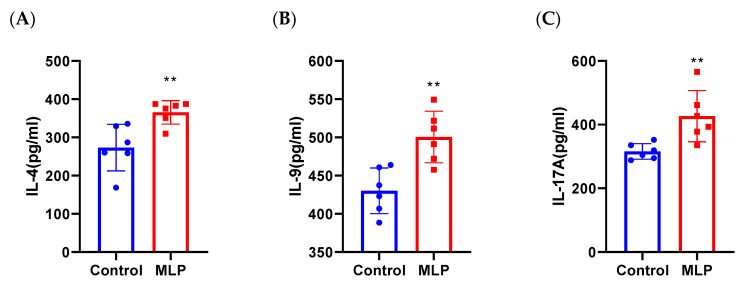
(**A**) Level of Th2 (IL-4), (**B**) Th9 (IL-9), and (**C**) Th17 (IL-17A)in MLP-sensitized mice splenocyte supernatant by ELISA. Significant difference (** *p* < 0.01; *n* ≥ 5) compared with the control group.

**Figure 6 nutrients-14-04700-f006:**
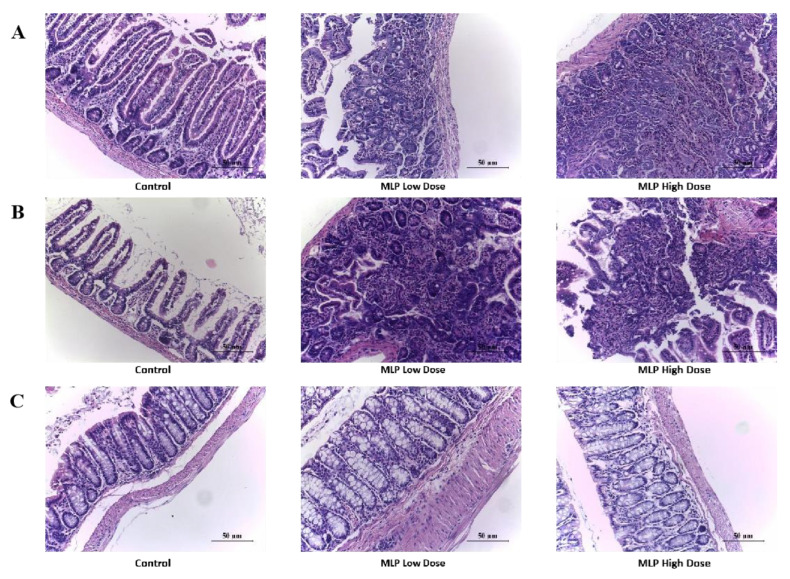
Histological sections (HE) of intestine ((**A**): jejunum; (**B**): ileum; (**C**): colon) tissues from the MLP high-dose group and MLP low-dose group.

**Table 1 nutrients-14-04700-t001:** Amino Acid Analysisof MLP.

Pk #	RT	Name	Height	Area	ESTD Conc/nmol	Units
1	4.7	Asp	135,861	2,135,172	0.155	mol/kg
2	5.34	Thr	90,990	1,474,391	0.099	mol/kg
3	5.927	Ser	124,707	2,056,568	0.135	mol/kg
4	6.66	Glu	618,119	11,656,396	0.809	mol/kg
6	9.44	Gly	258,741	4,983,018	0.348	mol/kg
7	10.26	Ala	125,102	3,034,497	0.216	mol/kg
8	11.54	Cys	54,848	775,330	0.05	mol/kg
9	12.193	Val	133,756	2,104,260	0.154	mol/kg
11	13.467	Met	51,020	1,200,018	0.071	mol/kg
13	15.747	Ile	55,246	1,503,985	0.111	mol/kg
14	16.947	Leu	97,766	2,778,678	0.226	mol/kg
15	17.767	Tyr	31,653	613,296	0.05	mol/kg
16	18.680	Phe	71,998	1,519,116	0.114	mol/kg
19	20.860	Lys	40,882	626,199	0.041	mol/kg
20	22.260	NH_3_	362,631	8,340,763	1.051	mol/kg
21	23.180	His	55,188	1,004,429	0.07	mol/kg
23	27.207	Arg	237,198	6,324,650	0.466	mol/kg

## Data Availability

The data that support the findings of this study are available from the corresponding author upon reasonable request.

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
