# Peer review of "Potential Allergenicity Response to Moringa oleifera Leaf Proteins in BALB/c Mice"

_nutrients, 2022, doi:10.3390/nu14214700_

Round 1

Reviewer 1 Report (Previous Reviewer 2)

I checked the MS and found interesting now. Before I reviewed it and that time comments has been properly endorsed by authors.

Author Response

Thank you so much for your review and guidance on our research!

Reviewer 2 Report (Previous Reviewer 1)

Zhang et al., propose and evaluated the allergenicity of Moringa oleifera leaves extract in an experimental model.

The manuscript needs to be check in regard of the typography, over the whole manuscript are spaces between words missing. The main focus of the paper is misspelled as moringa oleifera instead of Moringa oleifera.

Several references are missing in the whole manuscript, especially in the methods section, where only 4 reference where used.

The animal model (Brandt et al., 2003) used as a base in this manuscript, differs greatly from the current model, probably the differences should be indicated, as Brandt used OVA and the difference in the concentration.

Was the score system done in a blinded way?. Could the different aspects of the score be shown? Not only the final score?. It would be interest to see the different effect. Its stated that the effect were from mild to lethal, how many mice died during the experiment?

The term food allergy should be used carefully, as the model drom Brandt is crearly known as stated as an allergic diarrhea model.  Having this in consideration, in lines 165-166 a diarrhea assesment was mention, but not shown. Due to the fact that the model is indeed and allegic diarrhea model, this data should be provided.

The ELISAs need to be shortly described, especially because the cited references do not use Moringa, but known allergens. This discrepancy between using single allergens or extracts may explain the extreamly high OD values shown in the IgE and IgG results. This values are completely not feasible. First, because the linear limit if the device used at 450 is from 0-4 and there are even higher OD reported. Second, if the mice in the control group never consumed Moringa (as stated in line 142), how can the high level of sIgE (aprox 3) be explained?. If that would be the case and a valid result, this will be reflected in the immunoblot where high IgE content would regognize the extract in the negative control. I risk to speculate that there is a extrem high background in the ELISA that need to be fix before publishing the data. Marsteller used as reference for the method show ODs of 0.1 for the controls and 0.2 to the experimental groups, which ilustrates my point.

The figures and the leyends need to be improved. The figures are not at all in a consecutive or logic number (1,2,4,6,3,4,5).Figure 4 is repeated. Figure 2 only indicates subpart C and A is no a SDS-PAGE, its an immunoblot, there is no clear indication of what is depicted in the figure.

Immunoblot is missing in the methods. How it was done? Which Sera was used?

The mast cells were quantified? There is the indication that were significantly enhanced (309-310), but the data is shown only in a qualitatative way, how was this statistically analysed?. It would be importat to measure the mMCP-1 as mast cell activation marker as performed by Brandt et al.

The discussion needs to be inproved to analyse and discuss the results obtained, until now its a bibliografic description. For example what the whole data from Fig 2 means, beside that the extract showed reactivity (its not clear if its an IgE, IgG reactivity), I assume the sera from the sensitized mice was used.

Author Response

Dear editor Ms. Selena Liu and reviewer,

Thank you very muchfor giving us a chance to resubmit our rejected manuscript (Manuscript ID: nutrients-1985414; title: Potential allergenicity response to Moringa oleifera leaves proteins in BALB/c mice). We have revised the manuscript carefully according reviewer’s comments, and the revision was marked in red in the manuscript. We hope our resubmitted manuscript will be undergo peer review again.

The main corrections in the manuscript and the responds to reviewer’s comments are as follow.

Reviewer 2:

Comments to the Author

(1) The manuscript needs to be check in regard of the typography, over the whole manuscript are spaces between words missing. The main focus of the paper is misspelled as moringa oleifera instead of Moringa oleifera.

Response: We would like to thank you for your positive comments and suggestions. We have checked the typography and added the space between words missing. We have corrected the spelling errors as Moringa oleifera instead of moringa oleifera.

(2) Several references are missing in the whole manuscript, especially in the methods section, where only 4 reference where used.

Response: Thank you very much for your reminding. We have added 7 references in the methods section(line 118, 124, 128-129, 142,202,232,252), 4 references in the other section (line 274,464,465,471).

(3) The animal model (Brandt et al., 2003) used as a base in this manuscript, differs greatly from the current model, probably the differences should be indicated, as Brandt used OVA and the difference in the concentration.

Response: Thank you very much for your reminding. We have added the explain with "However, because the sensitizer is different from the reference, the dose was adjusted. For example, because MLP is a mixed protein, part of which may be the allergen, so the sensitization amount was 6-12 times higher than the 50ug of OVA, and the excitation amount was set at 5mg due to the solubility."(Line158-161)

(4) Was the score system done in a blinded way?. Could the different aspects of the score be shown? Not only the final score?. It would be interest to see the different effect. Its stated that the effect were from mild to lethal, how many mice died during the experiment?

Response: Thank you very much for your suggestion. The score system was done in a double blind method and we have added that in line 189-190. We have described the three aspects of the system score as " The systemic score (0–6) was defined as the sum of scores obtained from three criteria: scratching behaviour, loss of mobility and swelling. Scratching was graded based on the average number of scratching events during each 15-min interval (three intervals during a 45-min observation) as follows: 0,1–3 events, 1,4–5 events, 2,>6 events. Loss of mobility was graded according to immobility duration as follows: <10 min =0, >10 min=1, during the entire trial=2. Swelling (bristled fur, oedema around the nose and eyes, and laborious breathing) was graded as follows: none=0, swelling=2 " (Line190-197) and showed the score of the three aspects in Fig. 3A(Line 324). There were three mices died in the MLP high dose group in the experiement.

(5) The term food allergy should be used carefully, as the model drom Brandt is crearly known as stated as an allergic diarrhea model.  Having this in consideration, in lines 165-166 a diarrhea assesment was mention, but not shown. Due to the fact that the model is indeed and allegic diarrhea model, this data should be provided.

Response: Thank you very much for your review. We agree with your opinion and we have replaced some sentences with "To assess the sensitizing capacity of allergic reactions induced after the challenge with a heavy amount of proteins in sensitized mice, we studied the clinical symptoms. The potency of food allergy were determined by the assessment of the rectal temperature change , the systemic anaphylaxis symptoms score and the diarrhea score. " and "Diarrhea was assessed by visually monitoring mice for up to 1 hour following intragastric challenge. Diarrhea scored arbitrarily as follow: 0, no fecal changes; 1, soft but well-formed faces; 2, soft and non-formed faces; 3, one episode of liquid diarrhea; 4, at least two episodes of liquid diarrhea; 5, score 4 plus only clear liquid in the colon at the sacrifice(Elkholy et al., 2019). " Moreover, we have added the result of the diarrhea score in Fig. 3B and have added the sentences " Compared to the control group, the MLP group exhibited a significant elevation of the diarrheal score " (Line324-325) .

(6) The ELISAs need to be shortly described, especially because the cited references do not use Moringa, but known allergens. This discrepancy between using single allergens or extracts may explain the extreamly high OD values shown in the IgE and IgG results. This values are completely not feasible. First, because the linear limit if the device used at 450 is from 0-4 and there are even higher OD reported. Second, if the mice in the control group never consumed Moringa (as stated in line 142), how can the high level of sIgE (aprox 3) be explained?. If that would be the case and a valid result, this will be reflected in the immunoblot where high IgE content would regognize the extract in the negative control. I risk to speculate that there is a extrem high background in the ELISA that need to be fix before publishing the data. Marsteller used as reference for the method show ODs of 0.1 for the controls and 0.2 to the experimental groups, which ilustrates my point.

Response: Thank you very much for your review. We realized that there was really a big background in determining IgE and IgG by ELISA, so we have corrected the data and replaced the results in Fig. 3D,E(Line338,359).

(7) The figures and the leyends need to be improved. The figures are not at all in a consecutive or logic number (1,2,4,6,3,4,5).Figure 4 is repeated. Figure 2 only indicates subpart C and A is no a SDS-PAGE, its an immunoblot, there is no clear indication of what is depicted in the figure.

Response: Thank you very much for your careful review. The number confusion may be due to the format change after uploading. We have checked the number of the figure and pasted the figures in order. The sub part A of Figure 2 is to evaluate the molecular weight of each protein in MLP, using SDS-PAGE. Line 359.

(8) Immunoblot is missing in the methods. How it was done? Which Sera was used?

Response: Thank you very much for your suggestion. Immunoblotting would identify allergens, and subsequent experiments will further study MLP allergens. So we have depicted in the conclutions section one of the limitations of the experiement was "isolating pure proteins and performing immunoblotting with the serum of allergic patients" (Line 575-576) .

(9) The mast cells were quantified? There is the indication that were significantly enhanced (309-310), but the data is shown only in a qualitatative way, how was this statistically analysed?. It would be importat to measure the mMCP-1 as mast cell activation marker as performed by Brandt et al.

Response: Thank you very much for your suggestion. We randomly selected 20 visual fields of jejunum sections for counting the number of mast cells, and supplemented the quantitative data graph of mast cells in Fig. 3G(Line357,363-364). We agree with your opinion that the mMCP-1 as a marker of mast cell activation is very important, so it had been measured in the experiment. But the data showed no significant difference between the control group and the MLP group, so it was not included in the manuscript.

(10) The discussion needs to be improved to analyse and discuss the results obtained, until now its a bibliografic description. For example what the whole data from Fig 2 means, beside that the extract showed reactivity (its not clear if its an IgE, IgG reactivity), I assume the sera from the sensitized mice was used.

Response: Thank you very much for your suggestion. We have added the discussion about the date from Fig2 as follows: "As an important source of protein supplement, Moringa oleifera leaves have been in-troduced or imported to many countries that did not grow and eat Moringa oleifera leaves before because of their extremely high protein content and nutrients(Islam et al., 2021). Almost all edible plants have been reported to cause allergy(Zuidmeer et al., 2008). As a food rich in plant protein, MLP may contain allergen protein. In general, many allergens in plants are mainly composed of proteins with low molecular weight and disulfide bonds. According to these properties, we determined the molecular weight, three-dimensional structure and composition of the extracted MLP. At present, SDS-PAGE gel electrophoresis is mainly used to determine the molecular weight of allergens(Zhang et al., 2016). However, the relative mobility of some proteins in SDS system does not show a linear relationship with their molecular weights, so SDS-PAGE has some errors in the determination of protein molecular weights. In order to more accurately determine the molecular weight of MLP, we also used MALDI-TOF-MS method to accurately determine the molecular weight of protein subunits from Moringa oleifera leaves,which were shown in Figure 2. In order to better understand the properties of MLP used in this study, the protein structure and amino acid composition were identified by infrared spectroscopy and amino acid analysis. The structural information of the molecular weight of these proteins can also provide help for further functional identification of leaf proteins. This result will provide some basic information for better understanding the potential allergens of MLP."Line 462-481.

Nevertheless, we still believe our creative work will contribute to a new finding of allergen of food allergy. Our prospective study will provide strong evidence to expand the current understanding of potential risk factors for food allergy and suggest that moringa oleifera leaves proteins should be considered as a potential factor for food allergy. Accordingly, we would like to resubmit the manuscript and ask if you could offer us an opportunity to undergo peer review and receive comments from the reviewers again.

I declare that the work described here is original work that has not been published anywhere. We are looking forward to getting a positive response at your earliest convenience.

Sincerely,

Hongbing Chen, corresponding author

Professor in Food Science, Nanchang University

State Key Laboratory of Food Science and Technology

E-mail: chenhongbing@ncu.edu.cn

Round 2

Reviewer 2 Report (Previous Reviewer 1)

The manuscript by Zhang et al., include several modifications and it’s improved, I congratulate the authors for the effort on it. However, there are still some points remaining:

1. There are still a lot of typos in the manuscript including a lot of missing spaces. Just to illustrate my point, I identified 14 typos in 30 lines (between lines 33 to 63). Between lines 222-226 there are not points separating sentences. This occur during the entire paper including the graphs axis. I strongly recommend a professional English editing.

Examples:

Line 20: (MLP)_have

Line 26: MLP._This

Line 33: Asthma instead of asthmaina

Line 36: France_ (Poussel)

Line 38: asthma_(OA),_a

2. In the introduction are mentioned previous studies in the lab with Moringa, but are not cited in the manuscript.

3. Fig 2 legend is still missing information. Section A its clearly a SDS-PAGE. It will be important that the legend indicate to the readers what is depicted in the figure in each line of the SDS-PAGE.

4. I am wondering that 2 different ELISA readers where using depending on the antibody/cytokine measured. Is a reason behind it?

5. Figure 3. is improved, but the ODs associated to sIgE are still quite high in the mice that never consumed Moringa. This is extremely unusual, especially when the ODs are even higher than the IgG, which normally is 10 times higher. Can other method be used to measure the IgE? Probably the ELISA needs further characterization and standardization.

6. In the conclusions it is stated that this is an anaphylactic model, but it was discussed before as a food allergy model and by the original reference as an allergic diarrhea model. Could you please clarify exactly which model this is?

Author Response

Dear editor Ms. Selena Liu and reviewer,

Thank you very much for giving us a chance to resubmit our rejected manuscript (Manuscript ID: nutrients-1985414; title: Potential allergenicity response to Moringa oleifera leaves proteins in BALB/c mice). We have revised the manuscript carefully according reviewer’s comments, and the revision was marked in red in the manuscript. We hope our resubmitted manuscript will be undergo peer review again.

The main corrections in the manuscript and the responds to reviewer’s comments are as follow.

Reviewer 2:

Comments to the Author

  1. There are still a lot of typos in the manuscript including a lot of missing spaces. Just to illustrate my point, I identified 14 typos in 30 lines (between lines 33 to 63). Between lines 222-226 there are not points separating sentences. This occur during the entire paper including the graphs axis. I strongly recommend a professional English editing.

Response: Thank you very much for your review. We have specially invited Dr. Chunfu Zheng to edit this article. Dr. Chunfu Zheng is an Adjunct Professor at the University of Calgary, Calgary, Canada.

  1. In the introduction are mentioned previous studies in the lab with Moringa, but are not cited in the manuscript.

Response: We thank you very much for your reminding. Because of the previous research results in our laboratory which mentioned in the introduction section ( unpublished data ), there are no references cited in this manuscript.

  1. Fig 2 legend is still missing information. Section A its clearly a SDS-PAGE. It will be important that the legend indicate to the readers what is depicted in the figure in each line of the SDS-PAGE.

Response: We thank you very much for your reminding. Based on your suggestion, we have re-examined and revised the Figure 2 legend.(Fig2, Line 265).

  1. I am wondering that 2 different ELISA readers where using depending on the antibody/cytokine measured. Is a reason behind it?

Response: Thank you very much for your careful review. We found our description in the determination of IgE and IgG was repeated and confused, which depicted in 2.4.4 and 2.4.5 section. The use of two different ELISH readers was not intentional in that the determination of IgE and IgG was performed in different laboratories than other cytokine determinations, the instruments of each laboratory were used. But the IgE and IgG were measured using one ELISA reader as depicted in 2.4.4 section. So we have deleted IgE and IgG in 2.4.5 section (Line 227-231).

  1. Figure 3. is improved, but the ODs associated to sIgE are still quite high in the mice that never consumed Moringa. This is extremely unusual, especially when the ODs are even higher than the IgG, which normally is 10 times higher. Can other method be used to measure the IgE? Probably the ELISA needs further characterization and standardization.

Response: Thank you very much for your careful review. We agree with your opinion. The reason of ODs of the sIgE was higher than the sIgG1 in our results might be the different dilution ratio of IgE antibody and sIgG1 antibody. The IgE antibody is 1:200 (v:v) and the IgG antibody is 1:1000 (v:v) respectively. On the other hand, specific IgE and IgG1 levels were estimated including protein coating, serum and antibody incubation. There are many steps, which is easy to cause high background. The method used in our experiment was relative ODs measurement, so the background may have some influence.

  1. In the conclusions it is stated that this is an anaphylactic model, but it was discussed before as a food allergy model and by the original reference as an allergic diarrhea model. Could you please clarify exactly which model this is?

Response: Thank you very much for your suggestion. We have changed it and the new sentence is " To that end, we first developed a clinically relevant mouse model of allergic diarrhea, for which IgE is the main mediator and in which challenge is performed by oral administration of allergens, similar to what occurs in humans."(Line524)

Sincerely,

Hongbing Chen, corresponding author

Professor in Food Science, Nanchang University

State Key Laboratory of Food Science and Technology

E-mail: chenhongbing@ncu.edu.cn

This manuscript is a resubmission of an earlier submission. The following is a list of the peer review reports and author responses from that submission.

Round 1

Reviewer 1 Report

Zhang and collaborators evaluated the possible allergenicity to Moringa leaf extract in a mouse model as an alternative to the lack of human studies.

The manuscript need major improvement, there is a big lack of references all over the text, continuesly spaces are missing in the text and the way that BALB/c is written shoulf be corrected in several parts of the text including the title.

There is no clear indication of a recognition of the protein of the extract by the mice, which could be done with an immunoblot for example.

The figures need to be improved, specially figure legends. Figure 1 indiccates an SDS-PAGE, but not information of what is depicted in the blot. section C is missing.

There is not a minimum indication of which proteins are recogized by the mice in the extract and their identification.

It is a complete lack of discussion about the "anti-allergic" and anti-histaminic effects reported for Moringa.

While high titers of IgE and IgG are shown in a ELISA, it calls the attention that the ODs are extreamly high for an ELISA, including the mice  in the negative control (OD around 3).

Reviewer 2 Report

I have evaluated the manuscript entitled “Potential allergenicity response to Morenga leaf Protein in Balb/C mice” by  Ji Zhang et al, and the author has done a great job by executing the novel aspect of Morenga leaf. I found this article interesting for the readers and follow the scope of the journal. I don’t have any major comments as this review article is well organized, however, the author needs to elaborate the discussion of results with collected data, and the conclusion should be included with expert comments and a path forward. There are technical errors, and grammatical errors, I hope the editor will take care of them. Some of the technical errors are below which needs rectifications:

In the abstract section line 1,2 needs rephrasing by replacing the words anti oxidation by antioxidants, blood sugar by blood glucose ,blood fats by blood cholesterol. Also line 3 needs revision as some people shows allergic reactions to Morenga leaf????? Line 4 also needs revisions as limited studies were conducted. Remove coma after IgG.

In materials in methods section please add a little bit regarding the feed used by the mice, as to define that the allergic reaction was due to the feed  given to mice or due to Morenga leaf. Better to add a sketch to the experimental study design for easy readers understanding.

In section 2..4.7 You mentioned that after cutting the sections and preparation of slides you stained by using Haematoxylin and eosin and Toludine blue stain. What was the purpose of addition Toludine blue stain after H and E staining????? Please elaborate and justify yours statement.

What was yours matrix of grading inflammation??Gave reference?????

You calculated the mast cells under light olumpus microscope. Please mention the magnification power under which you counted??And what was yours method of counting?

Add limitations of yours study.

Reviewer 3 Report

The authors addressed an interesting subject, and the study is wee designed and written in good English. The article could be published after minor revisions;

Title: Since the gender of mice may affect the immune response, please consider adding the gender to the title. 

1- There are minor errors in the writing and typing of the article (although the language is scientific and sounds good). For instance, in the abstract, it is written that "Enhanced levels of histamine and Th-related cytokines releasewere observed in the sensitized mice serum.". Here, the space between release and were is missing, which should be revised. there are some other minor errors like this one, which should be revised.

2- In the methods;

(a) Since the time and dose of administered MLPs affect the sensitization, please provide enough information (or citation) on how the researchers decided to follow the mentioned experimental procedures?

(b) The researchers have utilized two different doses (high and low), while it is suggested to use different serials of the MLP. How did the researchers decide on that? 

(c) Figure 1. Mouse immunization protocol does not have a proper quality and does not present enough information. Please consider revising it. 

(d) Regarding the animal studies, did you follow all ethical regulations?

3- Results:

(a) in figure 2, the information on the graphs is not thorough, for instance in figure 2A, the score should be defined on the graph. Moreover, it is suggested to change the OD format in IgG and IgE to UI/mL.

(b) Figure 5 is not informative. Please enrich the legends, and show the morphological changes by arrows. 

Discussion;

(a) The authors have not discussed the possible molecular mechanisms that might be associated with inducing allergy. Please enrich the discussion. 

(b) What were the limitations of the research? Please discuss.